# Improvement of Delegated Proof of Stake Consensus Mechanism Based on Vague Set and Node Impact Factor

**DOI:** 10.3390/e24081013

**Published:** 2022-07-22

**Authors:** Runyu Chen, Lunwen Wang, Rangang Zhu

**Affiliations:** College of Electronic Engineering, National University of Defense Technology, Hefei 230037, China; chenrunyu17@nudt.edu.cn (R.C.); zhurangang17@nudt.edu.cn (R.Z.)

**Keywords:** blockchain, consensus mechanism, DPoS, vague set, fuzzy value, impact factor

## Abstract

The Delegated Proof of Stake (DPoS) consensus mechanism uses the power of stakeholders to not only vote in a fair and democratic way to solve a consensus problem, but also reduce resource waste to a certain extent. However, the fixed number of member nodes and single voting type will affect the security of the whole system. In order to reduce the negative impact of the above problems, a new consensus algorithm based on vague set and node impact factors is proposed. We first use fuzzy values to calculate the ratings of all nodes and initially determine the number of agent nodes according to the preset threshold value. Then, we judge whether a secondary screening is needed. If needed, calculating the nodes’ impact factor based on their neighboring nodes, and combining their impact factors with adjacency votes to further distinguish the nodes with the same fuzzy value. In addition, we analyze the dynamic changes in the composition and scale of the agent node set and give its ideal size through testing. Finally, we compare the proposed algorithm with DPoS algorithm and existing fuzzy set-based algorithms in different scales and network structures. Results show that no matter in what kind of network structures, the effectiveness of the proposed algorithm is improved. Among which, the most noticeable improvement is seen in complex network structures.

## 1. Introduction

Blockchain technology originated from Bitcoin [1]. It is a model for implementing and managing the processing of works in a peer-to-peer networking (P2P networking, also known as peer-to-peer network) environment by constructing blockchain data structures that cannot be forged, tampered with, and traced through transparent and trustworthy rules [2]. In the Internet information era, information on the network is open and transparent, but it cannot be fully trusted because of artificial and arbitrary tampering. Blockchain is widely used in various fields such as privacy [3], public cloud storage [4], edge computing [5], etc. [6] because of its decentralization, immutability, and is de-trusted, which give data on the Internet a new value that can be trusted again.

The consensus mechanism is the core technology of blockchain and has been studied by many researchers [7,8,9]. The continuous development of blockchain has also contributed to the diversity of consensus mechanisms. Currently, there are four main consensus mechanisms in common use: Proof of Work (PoW), Proof of Stake (PoS), Delegated Proof of Stake (DPoS), and Practical Byzantine Fault Tolerance (PBFT).

PoW [10] was first proposed by Dwork to specifically deal with junk mail and control access to shared resources. Later, Satoshi Nakamoto used PoW to ensure the consistency of the Bitcoin system. Although PoW has simple logic, easy implementation, complete decentralization, and high security, its system efficiency is low, the consensus period is long, and its mining mechanism will cause a lot of resource waste, which is not suitable for commercial applications.

Sunny proposed the PoS mechanism in 2012 [11]. Compared with PoW, PoS solved the disadvantages of resource waste and arithmetic concentration and shortened the time to reach consensus to a certain extent, but PoS was prone to bifurcation, and its security and fault tolerance have not been strictly mathematically proven. Therefore, Larimer proposed DPoS, which uses the witness mechanism to convert the bookkeepers in the PoS consensus into a small group composed of a specified number of nodes [12]. These witnesses are elected through a decentralized voting mechanism, which ensures the democratization of the election. It also reduces the requirements for confirmation and realizes consensus verification at the second level by trusting a small number of honest nodes. In addition, DPoS is regulated and has good performance in effectiveness, resource consumption, and fault tolerance. However, the implementation of DPoS is relatively complex and has the phenomenon of a “weak center”.

This paper starts with the voting process of the DPoS consensus algorithm, and proposes a new consensus algorithm by combining vague sets and the impact factors of the nodes themselves. The main improvements are as follows:We not only use the fuzzy set method to calculate the voting rate of all nodes, but also preset the threshold value of the voting rate according to all possible results of voting, so as to achieve the optimization of the screening results.We fully consider the different identities of all nodes in the system and the situation that multiple nodes may have the same vote rate in (1), and propose a calculation formula based on the influence factor of the node itself, and further distinguish nodes with the same vote rate by comparing the actual voting situation corresponding to their first-order and second-order adjacent nodes.We comprehensively consider the dynamic changes in the scale and internal composition of the member node set. By regularly replacing the member nodes, the risk of the system being attacked is reduced and the security of the algorithm is improved.

## 2. Backgrounds and Related Works

### 2.1. Fuzzy Value and Vague Set Theory

Zadeh proposed fuzzy set theory [13], where a fuzzy set F can be represented by $\left\{\left(u_{1},\mu_{F}\left(u_{1}\right)\right),...,\left(u_{n},\mu_{F}\left(u_{n}\right)\right)\right\}$, where u is the object in the set, and $\mu_{F}\left(u\right)$ is the membership function of fuzzy set F. The set of all objects is denoted by U. It is evident that $\forall u_{i}\in U,0\leq \mu_{F}\left(u_{i}\right)\leq 1$.

Gau and Buehrer proposed vague set [14] based on the fuzzy set, denoted by A in this paper. They defined the affiliation of a fuzzy set as the support, opposition, and uncertainty of the corresponding element, which describes some incomplete information clearer than the fuzzy set. Meanwhile, they split the affiliation into true affiliation $t_{A}\left(u\right)$ and false affiliation $f_{A}\left(u\right)$. Their relationship is as follows: $t_{A}\left(u\right),f_{A}\left(u\right)\subset \left[0,\;1\right],t_{A}\left(u\right)+f_{A}\left(u\right)\leq 1$. When the set U is a discrete set, a fuzzy set A can be expressed as:(1)$$A=\sum_{i=1}^{n}[t_{A}\left(u_{i}\right),1-f_{A}\left(u_{i}\right)]/u_{i}\;\;$$
where $\left[t_{A}\left(u_{i}\right),1-f_{A}\left(u_{i}\right)\right]$ represents the vague value of element $u_{i}$.

Xu et al. proposed a new DPoS consensus mechanism based on a vague set, similar to how human elections are held, and their method allows each node to vote for the agent node [15].

On this basis, many researchers have analyzed the relationship between a vague set and fuzzy set and proposed many ways to measure the fuzziness of a vague set (a vague set is a discrete set). Some of the more typical ones are the mathematical models proposed by Zhu [16] and Zhang [17]. Among these two models, Zhang thoroughly considered the connection and difference between the vague set and fuzzy set and proposed a mathematical model to convert a vague set into a fuzzy set.
(2)$$\mu_{A^{F}}=t_{A}\left(u\right)+\frac{1}{2}\left[1+\frac{t_{A}\left(u\right)-f_{A}\left(u\right)}{t_{A}\left(u\right)+f_{A}\left(u\right)+2\alpha }\right]\times \left[1-t_{A}\left(u\right)-f_{A}\left(u\right)\right],\alpha >0$$
where $\mu_{A^{F}}$ is the fuzzy affiliation function that can convert the element u in vague set A to the element in the corresponding fuzzy set $A^{F}$. α is a constant number, generally taken α=1. Equation (2) shows that when there are more favor votes than against, those who abstain from voting are more likely to choose in favor votes rather than against votes, and when there are more votes against than for, those who abstain from voting are more likely to vote against.

### 2.2. Related Works

Since the voting method has a certain degree of democracy, most of the existing consensus algorithms elect the agent nodes by voting. The researchers also made many improvements to the consensus algorithm voting process: Gao and Tong respectively combine EigenTrust [18] and Peertrust [19] reputation models with the voting process; Zheng uses the C4.5 decision tree [20] to calculate the nodes’ reputation; Wang introduces the concept of “credit” [21] and uses this index to decide the agent nodes. All the algorithms listed above solve the uncertainty caused by only counting votes to the algorithm. However, these algorithms more or less need to define weight parameters, and a parametric sensitivity analysis is necessary. The results can lead to a variety of interpretations. Unfortunately, most of these methods did not give a discussion of these parameters, but directly give a specific value. Therefore, there are certain limitations. Additionally, although these algorithms can ensure the honesty of the elected committee nodes to a large extent, with the accumulation of time, the reputation points of the nodes will tend to be centralized. This will lead to a centralization trend of the system.

Zhu noticed the emergence of the centralization trend, and proposed an algorithm based on dynamic clustering [22]. By setting up various mechanisms, this method largely avoids the emergence of the trend of system centralization under the premise of verifying the validity of the algorithm. However, the algorithm still only stayed on the positive part of the votes of the computing nodes while neglecting the negative part and the possible neutral part.

Xu thoroughly considered the three cases of affirmative, abstention, and negative votes in voting and combined vague sets [23] with the DPoS consensus mechanism for the first time. She demonstrated through experiments that the improved algorithm could distinguish the cases where multiple nodes get the same affirmative votes more effectively and summarized the probability distribution of tie-breaking votes. Based on this algorithm, Liu optimized the voting method by introducing the concept of adjacency voting [24], which enabled fewer nodes to participate in voting and made the whole system adjust the number of agents properly according to its size.

However, the algorithm proposed by Liu is deficient in three aspects: first, it ignored the prerequisite of using fuzzy sets. Taking the 10-node blockchain system as an example, as shown in Figure 1a, Liu assumed that the topology of the system is a circle so that the number of neighboring nodes of all nodes is 2, but the existing blockchain system is shown in Figure 1b, and the number of neighboring nodes of different nodes is not necessarily the same; second, the algorithm has the phenomenon of nodes giving themselves abstention or negative votes, which is contrary to the reality; third, the lottery algorithm is used to randomly determine agents when multiple nodes have the same fuzzy value, which may bring a certain amount of serendipity to the final result.

The algorithm proposed in this paper aims to solve the problems listed above.

## 3. Overview of the Improved Algorithm

The flow chart of the algorithm is shown in Figure 2. It can mainly be divided into 3 steps.

Voting, calculating the fuzzy value of all nodes, and conducting the first-round screening according to the threshold λ;Judging the impact factor $I_{i}$, deciding whether a second-round screening is needed, and how to conduct it according to the result of the first-round screening;Adjusting the number and composition of the set of agent nodes adequately based on the total number of nodes in the network.

Assuming that all nodes have the same identity, this paper takes the 12-node model shown in Figure 3 as an example and details all possible scenarios in the algorithm.

### 3.1. Fuzzy Value and Its Threshold

Define $V_{0}$ as the voting matrix, $V_{ij}$ represents the vote from node i to node j, where:$$V_{ij}=\left\{\begin{array}{cc} 1 & \mathrm{favor vote}\\ 0 & \mathrm{absention vote}\\ -1 & \mathrm{against vote} \end{array}\right.$$

All nodes can vote for, abstain from, or oppose other nodes during the voting process, and by default, all nodes will vote for themselves. Here is one of the voting matrices:


$$V_{0}=\left(\begin{array}{cccccccccccc} 1 & -1 & 0 & 1 & -1 & -1 & 0 & -1 & 1 & -1 & -1 & 1\\ 0 & 1 & 1 & 0 & 0 & 0 & 1 & 1 & -1 & 1 & 1 & 1\\ -1 & 0 & 1 & 0 & -1 & -1 & 0 & -1 & 1 & 0 & 1 & 1\\ -1 & 1 & 1 & 1 & 0 & -1 & 0 & -1 & 1 & -1 & 0 & -1\\ -1 & 1 & 1 & -1 & 1 & -1 & 1 & 1 & -1 & 0 & -1 & 0\\ 0 & 1 & 1 & -1 & -1 & 1 & 0 & 0 & 0 & 0 & 1 & 1\\ 0 & 1 & -1 & 0 & 0 & 1 & 1 & 1 & 1 & 0 & 0 & 1\\ 0 & 1 & 1 & -1 & 0 & 0 & -1 & 1 & 0 & -1 & -1 & 0\\ 0 & -1 & 0 & 0 & 0 & -1 & 1 & 0 & 1 & 1 & 1 & 0\\ -1 & 0 & 0 & 0 & -1 & 0 & -1 & -1 & -1 & 1 & -1 & -1\\ 1 & -1 & -1 & 0 & 0 & -1 & -1 & -1 & 0 & 1 & 1 & 1\\ -1 & -1 & -1 & -1 & 0 & 1 & -1 & 0 & 1 & 1 & 0 & 1 \end{array}\right)$$


Assuming that the total number of nodes in the system is N, we improve Equation (2) in this paper, and the equation for calculating the fuzzy value is as follows.
(3)$$\mu_{A^{F}}{}^{'}=\frac{Nt_{A}\left(u\right)-1}{N-1}+\frac{1}{2}\left[1+\frac{\frac{Nt_{A}\left(u\right)-1}{N-1}-\frac{Nf_{A}\left(u\right)}{N-1}}{\frac{Nt_{A}\left(u\right)-1}{N-1}+\frac{Nf_{A}\left(u\right)}{N-1}+2\alpha }\right]\times \left[1-\frac{Nt_{A}\left(u\right)-1}{N-1}-\frac{Nf_{A}\left(u\right)}{N-1}\right]$$

Compared with Equations (2) and (3) does not consider the node’s own favorable votes. Here are the reasons: firstly, counting the votes of nodes without considering the node’s own favorable votes can improve the objectivity of the voting result; secondly, we need to calculate the impact factor of each node in this paper, and counting the node’s own favorable votes will make the screening results more biased towards those with higher impact factor so that fairness cannot be guaranteed.

Taking node 2 as an example, the second row of the matrix shows that node 2 received a total of 7 favor votes, 4 abstention votes and 1 against vote. Therefore, according to the definition of fuzziness, we can calculate: $t_{2}\left(u\right)=0.545$, $f_{2}\left(u\right)=0.091$, so $\mu_{A}{}^{F}\left(2\right)=0.7586$.

After using the method above to calculate the fuzzy value corresponding to all nodes, the nodes are first arranged in descending order according to their fuzzy value, and in the case of the same fuzzy value, the nodes are arranged in descending order according to the number of favorable votes they received, with results shown in Table 1.

From Table 1, we can see that using fuzzy values to calculate the votes of nodes has a more intuitive differentiation of nodes’ voting situation than the traditional method of counting the number of votes. Meanwhile, adding the abstention vote makes the voting results more objective and closer to reality and can filter out the nodes corresponding to the conditions more effectively.

Next, we will discuss how to determine the committee node. Most of the traditional algorithms select them one by one according to one specific index, and this will lead to a high possibility that some of the agent nodes only obtain less than half the support rate, which will bring risks to the whole system. In real-life voting activities, not only the number of votes received is considered, but sometimes additional provisions are made for the vote percentage to ensure the reliability of the results. Therefore, we set a threshold value λ on their fuzzy value to conduct first-round screening. It must satisfy the following two conditions:
(1)Against votes must be less than one-third of the total votes (excluding its own vote);(2)The number of favor votes received by a node (excluding its own vote) must be higher than the number of against votes.

Therefore, if a system has N nodes, λ can be calculated according to Equation (4).
(4)$$\lambda =\frac{\left[\frac{N-1}{3}\right]+1}{N-1}+\frac{1}{2}\left[1+\frac{\frac{\left[\frac{N-1}{3}\right]+1}{N-1}-\frac{\left[\frac{N-1}{3}\right]}{N-1}}{\frac{\left[\frac{N-1}{3}\right]+1}{N-1}+\frac{\left[\frac{N-1}{3}\right]}{N-1}+2\alpha }\right]\times \left[1-\frac{\left[\frac{N-1}{3}\right]+1}{N-1}-\frac{\left[\frac{N-1}{3}\right]}{N-1}\right]\;$$

In this paper, λ=0.5517, so $Y=\left\{2,7,6,9\right\}$. Compared with the traditional algorithms, the threshold setting may increase the complexity of the whole algorithm. However, it can ensure that all screened nodes have a high support rate so that the system’s security can be enhanced. The flow of the first round of screening is shown in Algorithm 1.
**Algorithm 1:** First-round screeningInput: node i, fuzzy value $\mu_{A}{}^{F}\left(i\right)$, threshold λ
Output: alternative set Y
1: sort by $\mu_{A}{}^{F}\left(i\right)$ in descending order2:   **if** $\mu_{A}{}^{F}\left(i\right)\geq \lambda$ **then**3:     add i to Y
4:   **end if**5: **return** Y


Since all nodes in the system will vote based on the historical behavior of the destination node, we can roughly think that the vote rate is the reliability of the node. In addition, in the algorithm proposed in this paper, all non-member nodes can only perform the operation of receiving member node information. Therefore, once a node has an information mismatch, we can default it to a malicious node.

When the following situations occur, the entire voting process is invalid and a new vote is required:**Abstaining from voting contains more than half of the total number of votes.** In general, the votes of all nodes are based on the good or bad historical behavior of other nodes. Therefore, in this case, taking any node in the system as an example, the final votes of the node may only have the following two situations: in favor votes more than abstentions and negative votes, and negative votes more than votes in favor and abstentions votes. Therefore, under normal circumstances, it is almost impossible for the abstention votes cast by nodes in the system to exceed more than half of the total votes. Once it happens, we first need to observe not only the vote type of each node but also the vote type of each node to preliminarily determine the malicious nodes in the system, and then, after determining the malicious nodes, we need to re-vote.**All**$\mu_{i^{F}}$**are smaller than the threshold value**λ**.** Through simulation experiments and calculations, we found that this situation is often accompanied by the above-mentioned situations. Therefore, the response to this situation is the same as above.

### 3.2. Node Impact Factor

In measuring the importance of nodes in the system, Zhu used a way to determine the degree centrality of nodes based on node itself and its neighbor layer information [25]. Similar to this, we propose an easier method to determine the impact factor of different nodes in the network. We count the number of first-order and second-order neighboring nodes, and then assign the parameter 1, 0, 5 according to the relevant rules of information transmission.

Define i,j represents the node number, $k_{i}$ is the first-order centrality of the node, the union of all nodes constituted by U, the set of all first-order neighboring node numbers is $K_{i}$, the set of all second-order neighboring node numbers is $Z_{i}$, and the impact factor of the node $I_{i}$ is:(5)$$I_{i}=\sum_{j\in K_{i}}k_{i}+\frac{1}{2}\sum_{j\in Z_{i}}k_{i}$$

From Figure 3, we can easily calculate the first-order centrality of all nodes, which is shown in Table 2.

Taking node 5 as an example, $K_{5}=\left\{3,4,\;6,\;8\right\}$, $Z_{5}=\left\{1,\;2,\;7,\;9,\;11\right\}$, so we can calculate $I_{5}=22$.

All nodes’ impact factor is shown in Table 3.

From Table 2, we can see that when judging the impact factor of a node only from its first-order centrality, nodes 3, 4, 5, 6, 8, and 11 have higher impact factors; if we consider the first-order and the second-order centrality of the node, nodes 3, 4, 5, 6, 8, and 9 have higher impact factors in the system. Comparing node 9 with 11 in Figure 3, we can reach the conclusion that the later method is closer to reality. Therefore, evaluating the influence of a node in the system cannot only consider the number of its neighboring nodes but should also consider the magnitude of its neighboring nodes’ influence in the system.

### 3.3. Second-Round Screening

When there is a tie vote among multiple nodes, most algorithms often use the traditional coin toss method to randomly determine the winning node. Although this algorithm can greatly reduce the communication complexity, the random election of the winning node not only lacks a certain theoretical basis, but also may increase the risk of the system. The new method proposed in this paper can solve the above problems very well. We consider that blockchain is a complex system. The position of different nodes, the number of their adjacent nodes and their roles are different to a certain extent. Meanwhile, from the point of information transmission, the evaluation of a node’s adjacent nodes is often more valuable. Therefore, we comprehensively consider the influence factor of the node and the adjacent nodes when conducting the second-round screening. Here are the details:

When comparing the number of elements in Y with the actual number of nodes L in agent set X, there exists the following three circumstances:

Circumstance 1: L=4

In this circumstance, X=Y. There is no need to make further judgments.

Circumstance 2: L<4

In this case, we need to do further screening of the elements in Y. If L=2, we can get $X=\left\{2,7\right\}$ by comparing their fuzzy value; if L=3, since the nodes numbered 6, 9 have the same fuzzy value, it is necessary to consider the impact factors of these two nodes and their first-order and second-order adjacency nodes votes for further screening. The specific process is as follows:

According to Figure 2 and the voting matrix $V_{0}$, the neighboring votes of nodes numbered 6,9 can be derived as shown in the following table:

Define node support rate R, first-order neighboring node support rate $R_{K}$, and the second-order neighboring node support rate $R_{Z}$. The calculation formula is as follows:(6)$$R_{i}=I_{i}\left(R_{K}+\frac{1}{2}R_{Z}\right)$$
where:(7)$$R_{K}=\sum_{j\in K_{i}}\frac{I_{j}}{\sum_{k\in U}I_{k}}V_{ji}$$
(8)$$R_{Z}=\sum_{j\in Z_{i}}\frac{I_{j}}{\sum_{k\in U}I_{k}}V_{ji}$$

Take node 6 as an example, and we can calculate from the data in Table 4 to get $R_{K}=0.0682$, $R_{Z}=-0.0498$ so that $R_{6}=0.9526$, and similarly $R_{9}=1.4724\;>\;R_{6}$. Therefore, we can get $X=\left\{2,7,9\right\}$.

Circumstance 3: L>4

In this case, most traditional algorithms select the nodes with the higher fuzzy value among the remaining nodes to fill the vacant slots. These methods are simple, but since the fuzzy value of the remaining nodes does not reach the threshold, even if they are selected, it still makes the subsequent consensus protocols riskier and more unreliable. To address this problem, we propose a new judgment method divided into two main steps.

Step1: Determine whether all the remaining nodes get more than 1/3 of the total votes (excluding the nodes’ votes); if so, turn to Step2;

Step2: Finalize the remaining elements of the agent node-set X based on the votes of the nodes in set Y for all nodes that satisfy the conditions of Step 1.

If L=6, we need to select 2 nodes from the remaining nodes to the fill set X, and according to Step1, we first find the nodes that satisfy the condition, which are $\left\{5,3,4,11,12,1\right\}$; then, we calculate the vote percentage of each node according to Equation (9) based on the vote of the nodes above, and the results are shown in the following table.
(9)$$P_{i}=\sum_{j\in Y}I_{j}V_{ji}$$

From Table 5, we can find that $P_{12}>P_{4}>P_{3}>P_{1}>P_{11}$. Therefore, when n=6, $X=\left\{2,7,9,12,4\right\}$. In addition, no matter what L is, we can only find one corresponding X. The flow of the second-round screening is shown in Algorithm 2.


**Algorithm 2:** Second-round screeningInput: node i, alternative set Y, fuzzy value $\mu_{A}{}^{F}\left(i\right)$, total number n, impact factor $I_{i}$, voting matrix V
Output: member node-set X
1: **if** the number of elements in Y larger than n **then**2:   sort all $\mu_{A}{}^{F}\left(i\right)$ in descending order and find the largest n values3:   **if** all these $\mu_{A}{}^{F}\left(i\right)$ are not the same **then**4:      add the corresponding i to X
5:   **else**
6:      make further comparison6:      add all corresponding i to X
7: **else**8:   combined $I_{i}$ with V for further comparison of some of the remaining nodes 9:   add all corresponding i to X
10: **return** X



### 3.4. Dynamic Change of Agent Node-Set

The dynamic change in set X includes two aspects: the dynamic change in the composition of the agent node-set and the dynamic change in the size of the agent node-set according to the total number of nodes in the network.

#### 3.4.1. Dynamic Changes in the Composition of the Agent Node-Set

The DPoS consensus mechanism specifies that all nodes in the system will be recounted at regular intervals based on the running time of its consensus protocol, which means that the set of agent nodes will be replaced at regular intervals. In this paper, we develop personalized dynamic adjustment rules based on the characteristics of member nodes:
(1)If all $\mu_{A}{}^{F}\left(i\right)$ in X are larger than λ, we use the method of half replacement to replace half the number of nodes in X randomly.(2)If there are $\mu_{A}{}^{F}\left(i\right)$ in X smaller than λ, all the nodes which $\mu_{A}{}^{F}\left(i\right)<\;\lambda$ will be replaced, and the remaining nodes are still replaced by the method of half replacement.

Although the half-replacement method increases the centralization tendency of the system to some extent, the nodes retained before and after the replacement can maintain the consistency of the system and the stability of the system operation at the early stage of the next phase and avoid the possible disorder of information timing caused by the replacement of all nodes.

#### 3.4.2. Dynamic Changes in the Size of the Agent Node-Set

The DPoS consensus mechanism always decides the size according to the application environment. For example, the EOS system with the DPoS + BFT consensus algorithm has 21 members, and the Asch system with the DPoS + PBFT consensus algorithm has 101 members. The total number of nodes in the system for public and coalition chains changes a lot. If the number of agent nodes is fixed, it will inevitably increase the workload of these nodes. Based on this, this paper defines L/T as the ratio of the number of agent nodes L to the total number of nodes N. It has the following properties:
(1)L=0 represents the absence of an agent node in the network, and accordingly, the entire blockchain system will lack nodes for packaging transactions or information. In this circumstance, the blockchain will not function properly.(2)L=T represents that all nodes in the whole system are member nodes, so the blockchain system is a public chain, which is truly decentralized. A more typical one is the PBFT consensus algorithm.(3)L=1 represents that there is only one agent node in the whole system, which means that this node handles all transactions or data packaging and block generation. It is a typical centralized system and contradicts the decentralized nature of blockchain.

In general, we have L/T<1 and the specific value should be determined by the size of the corresponding blockchain system which will be analyzed experimentally in Section 4.

## 4. Experiment Analysis

The scale and network topology of blockchain vary in different application areas. Therefore, in order to test the performance of the algorithm proposed in this paper, we first compare the performance of multiple voting methods in terms of tie-breaking rate and accuracy in basic network structures such as star, linear, circular, tree, and complex network structures. Then, we summarize both the advantages and disadvantages of this algorithm. In addition, we discuss the ideal number of the agent number in systems of different sizes and give its interval range.

### 4.1. Tie-Breaking Rate

We define “a tie vote” as different nodes having the same fuzzy value and “a tie-breaking rate” as the probability of the occurrence of the corresponding event. All nodes can vote randomly regardless of the network structure. Therefore, the tie-breaking rate of voting results depends on the type of network topology. Based on this, we simulate voting for networks of different sizes and repeat the experiment 100 times for each structure; the number of occurrences of no tie-breaking, two-node tie-breaking, and multi-node tie-breaking for each network type is counted, respectively. The experimental results are shown in Table 6 and Figure 4.

From Figure 4, we can find that when the traditional DPoS voting method is used, there is only a 1.2% chance that all nodes obtain the same number of affirmative votes in the actual total number of 700 simulations. They all occur in small systems, so the traditional voting mechanism of the DPoS algorithm is not good enough to ensure reliability. Therefore, the voting mechanism of the traditional DPoS algorithm does not ensure the trustworthiness of each selected node. By analyzing the data in Table 6 and the curves in Figure 4, we can reach the following conclusions when using the fuzzy value to determine the votes of nodes: the differentiation of fuzzy value will be gradually vague as the scale of the system increases, and the probability of multiple nodes tying votes increases, even when the number of nodes in the system is 4, and the probability of no tie-breaking rate fails to reach 50%. Furthermore, there is hardly any system of such a tiny size in real life. Therefore, if the final election is done simply according to the magnitude of the fuzzy value, it will inevitably cause high riskiness of the election results of the member nodes and threaten the security of the whole system.

Further analysis of the 700 voting is presented below.

### 4.2. Interval Range of Agent Nodes

We conduct several experiments on systems of different sizes and count the number of nodes with fuzzy value higher than λ in each experiment; results are shown in Figure 5. The y-axis in Figure 5 represents the probability that a second-round screening is not required.

According to the probability distribution curves of systems of different sizes in Figure 5, we can reach the conclusion that when $⎣\frac{N}{4}⎦\leq L\leq ⎣\frac{N}{2}⎦$, there is a higher probability that a second-round screening is not needed so that it can minimize the communication complexity of the system.

### 4.3. Accuracy

Accuracy refers to the probability of successfully distinguishing node trustworthiness in the case of node tie-breaking. In this paper, the performance of the voting mechanism used in DPoS, the mechanism based on fuzzy value and lottery algorithm, and the algorithms proposed in this paper are compared and tested in different types of networks $\left(N,4\leq N\leq 12\right)$ by simulating L nodes selected in different types of system structures with different numbers of nodes. The obtained results are as follows.

#### 4.3.1. Simple Structures

In this paper, simulation experiments are conducted for the star, linear, ring, and tree structures, respectively, and the experimental results are shown in Figure 6.

Combining the characteristics of each network structure, we make the following analysis of the experimental results in Figure 6:(1)From Figure 6a, we can see that as the number of nodes in the system keeps increasing, the performance of the proposed algorithm in this paper decreases mainly because, in star structures, nodes can be divided into two main categories which result in there being only two kinds of node impact factors. As the data in Table 6 shows, as the number of nodes increases, the probability of nodes having the same vote situation also increases. Therefore, neither the judgment by the impact factor of the node nor the voting of the nodes can achieve the effect of further differentiating the nodes;(2)Different from Figure 6a, the curve in Figure 6b shows an increasing trend mainly because the star structure can be seen as a combination of several tree structures. When the number of nodes is small, the impact factors in the star structure are similar. However, as the number of nodes increases, the tree structure becomes complex, and the difference in node impact factors becomes larger. Therefore, the algorithm will have better performance in large-scale systems;(3)From Figure 6c,d, we can see that the algorithm proposed in this paper has almost the same performance as the existing fuzzy value-based algorithm in the circular and linear structures. Because of the specificity of these two structures, all nodes in the two structures have precisely the same number of neighboring nodes, and the influence factors of the nodes are also precisely the same.

From the above analysis, we can draw preliminary conclusions that the algorithm’s performance proposed in this paper is related to the complexity of the network structure. The more complex the network structure is, the better the performance is.

In order to test the conclusion above, we add a central point similar to the star structure to the circular structure and the linear structure, respectively, as shown in Figure 7a,b and the results obtained are shown in Figure 8. When comparing Figure 8a,b with Figure 6c,d, we can see that the algorithm’s performance is improved after the central point is added.

#### 4.3.2. Complex Structures

From Figure 9, we can see that the proposed algorithm has an average of 94% probability of finalizing the practical set of member nodes, which improves the performance by 14% compared to the algorithm proposed by Xu. The main reason for this is that there is a certain probability that different nodes have the same fuzzy value when the traditional vague set is used to determine the member nodes, but in complex structures, the nodes are divided and the probability that each node has the same influence factor is small so through second-round screening the nodes can be further differentiated.

Figure 10 shows the comparative performance of the algorithms in this paper in different structures.

From Figure 10, we can conclude that the system’s complexity primarily determines the performance of the algorithm. The more complex the system, the more secure the elected member nodes can be guaranteed.

The complexity of the network structure can be mainly reflected in two aspects: the number of nodes in the network and the structure of the network. Next, we will discuss these two aspects based on Figure 6, Figure 7, Figure 8, Figure 9 and Figure 10:**The number of nodes.** Generally speaking, for a system with a certain network structure, the greater the number of nodes, the more complex the system is. Therefore, by comparing the experimental results of each graph horizontally, we can roughly conclude that in the same network structure, with the continuous increase of the number of nodes in the system, the performance of the algorithm has been improved to some extent. Because in the case of a certain network structure, with the continuous increase of the number of nodes with voting rights, using the method proposed in this paper will cause the probability of multiple nodes to have equal votes and will gradually decrease, there is a high probability that only one round of screening can be performed to finally determine the member node set. This not only reduces the communication complexity of the election, but also the more obvious distinction of the node reputation value can further ensure that the elected nodes will not affect the security of the system.**The structure of the network.** Generally speaking, in the case of a certain number of nodes in the network, the complexity of the network depends on its structure. By longitudinally comparing the experimental results of different types of network structures in Figure 10, it can be seen that the performance of the algorithm in complex structures is better than other simple structures. Because of the algorithm proposed in the article, the network structure does not directly affect the results of the first round of screening. However, the difference in structure will lead to different positions of each node and the number of adjacent nodes in the system, which will result in different node impact factors. In this case, even if multiple nodes have the same vote ratio in the first round of screening, the second round of screening can well screen out the remaining member nodes due to the different impact factors of the nodes.

### 4.4. Fairness

The method we propose contains two elections in total, in which the first round of elections is necessary, and in this round, all nodes have the same voting share, which is not affected by the node’s own influence factor, so it can ensure fairness. The second round of elections is not inevitable. It only appeared when the first round of elections failed to yield a valid result. At this time, in order to better ensure the security of the final winning node, according to the principle of information transmission, it is necessary to comprehensively consider the influence factor of the node itself and the voting situation of its adjacent nodes. Although the second round of elections nominally introduces weights in the voting of each node, in fact the fairness of all nodes is not destroyed.

## 5. Conclusions

This paper first uses the improved fuzzy value calculation formula to calculate the fuzzy value of each node and sets a threshold value to conduct the first-round screening. The improved method does not consider nodes’ own affirmative vote. Then, we use the voting situation of neighboring nodes and the nodes’ impact factors to further judge the nodes’ voting rate and eventually determine the set of agent nodes. In addition, we briefly discuss the two aspects of dynamic change rules of the set of agent nodes and then give the ideal size of the agent node-set through experiments. Finally, tests are conducted on different scales and different network structures. The experimental results show that the performance index of the algorithm in this paper improves about 8% on average compared with the traditional algorithms. Especially in complex structures, it has the best performance. All improvements revolve around improving the fairness and democracy of voting results and ensuring the security of the system. However, with the increasing scale of the network, the use of the algorithm will lead to an increase in computational complexity. In the future, we will focus on reducing the complexity of the calculation while ensuring accuracy.

## Figures and Tables

**Figure 1 entropy-24-01013-f001:**
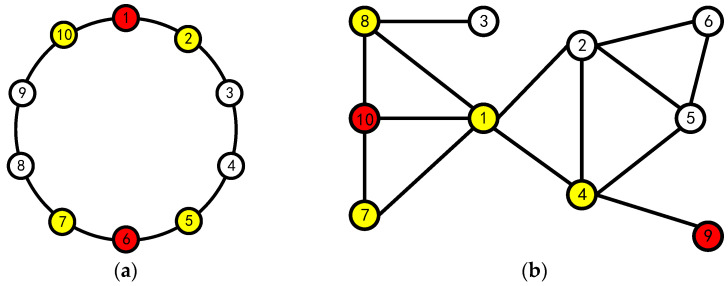
10 Node Topology (yellow nodes are the neighboring nodes of red nodes): (**a**) model diagram used by Liu and (**b**) actual node topology diagram.

**Figure 2 entropy-24-01013-f002:**
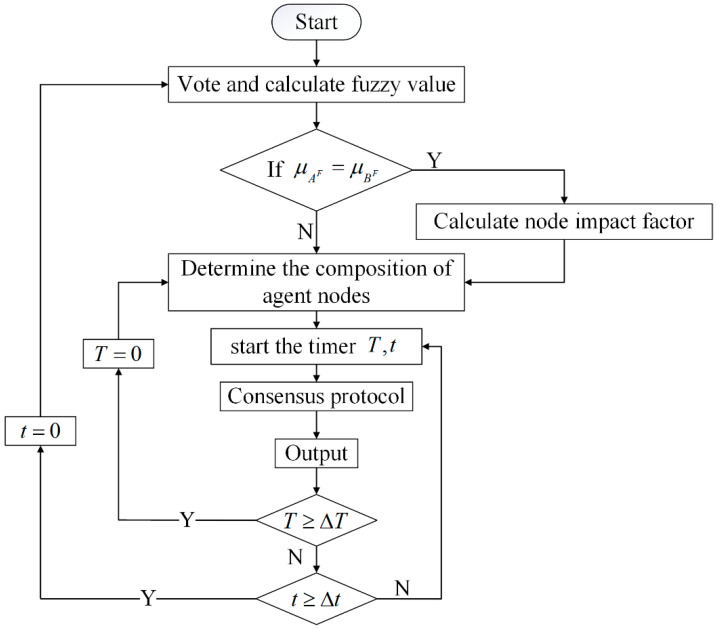
The flow chart of the algorithm. t,Δt dictate the time when the entire system will be reconfigured; T,ΔT dictate the time when the agent node set will be reconfigured; $\mu_{A^{F}},\mu_{B^{F}}$ dictate the nodes’ fuzzy value.

**Figure 3 entropy-24-01013-f003:**
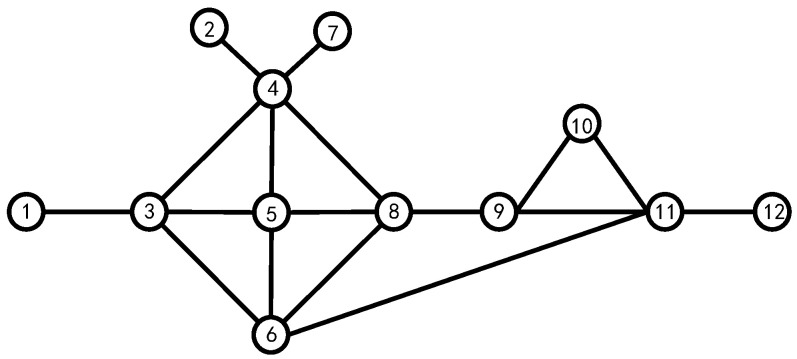
The 12-node model network.

**Figure 4 entropy-24-01013-f004:**
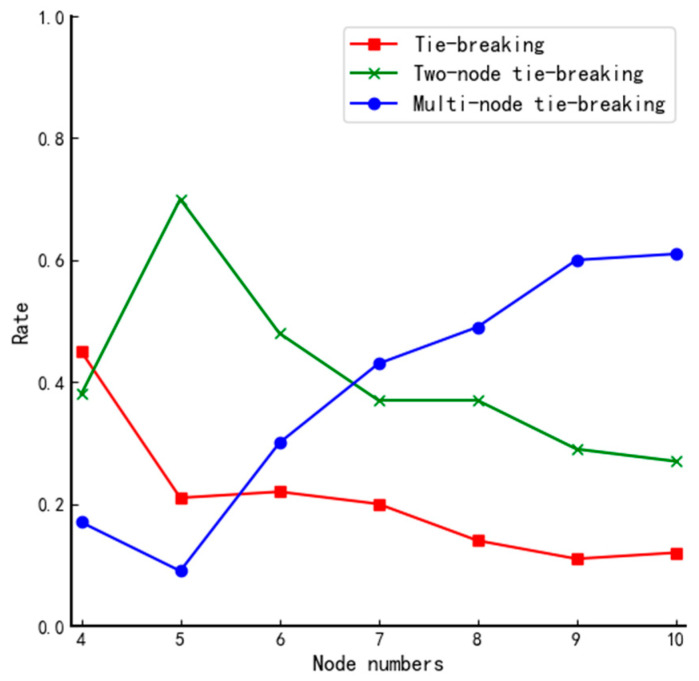
Comparison of tie-breaking rate for different size systems using DPoS.

**Figure 5 entropy-24-01013-f005:**
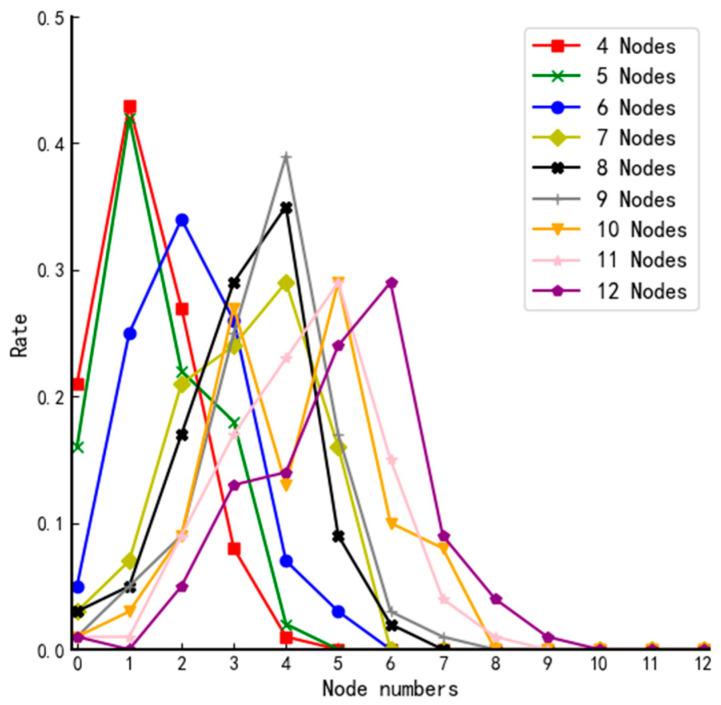
The probability distribution curves of systems of different sizes.

**Figure 6 entropy-24-01013-f006:**
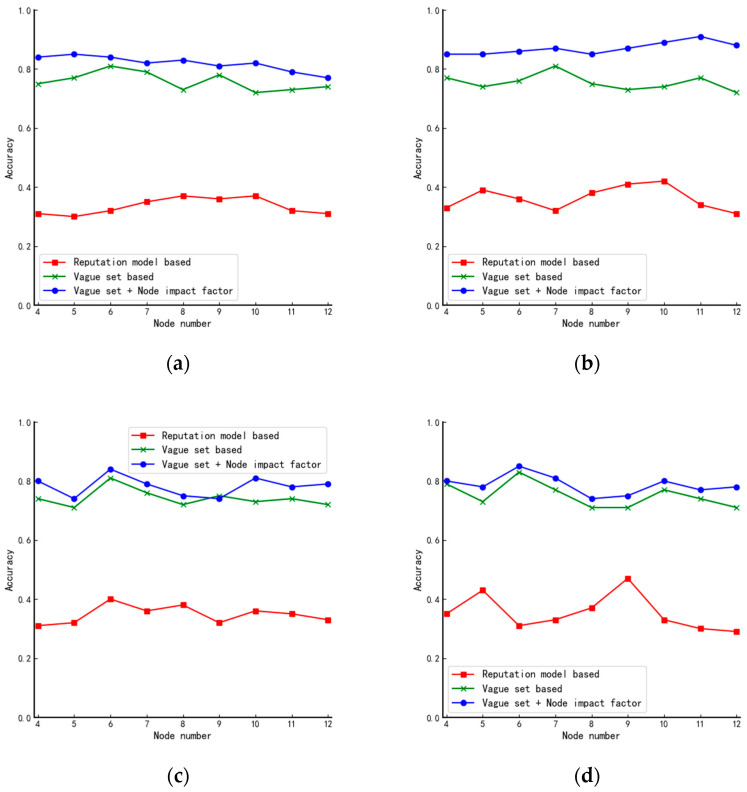
Comparison of accuracy of algorithms in (**a**) star, (**b**) tree, (**c**) circular, and (**d**) linear structures.

**Figure 7 entropy-24-01013-f007:**
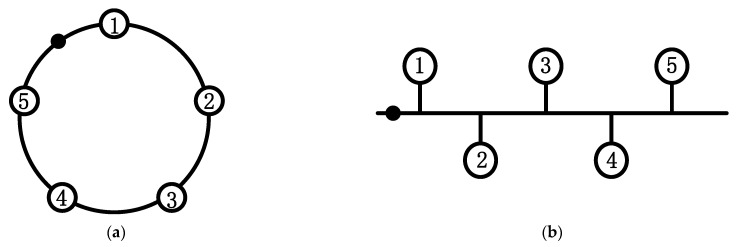
Structure after artificially adding the center point: (**a**) improved circular structure and (**b**) Improved linear structure.

**Figure 8 entropy-24-01013-f008:**
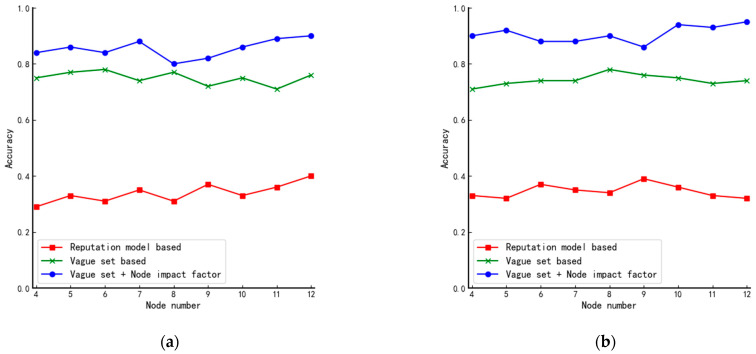
Comparison of accuracy of algorithms in improved structures: (**a**) improved circular structure and (**b**) improved linear structure.

**Figure 9 entropy-24-01013-f009:**
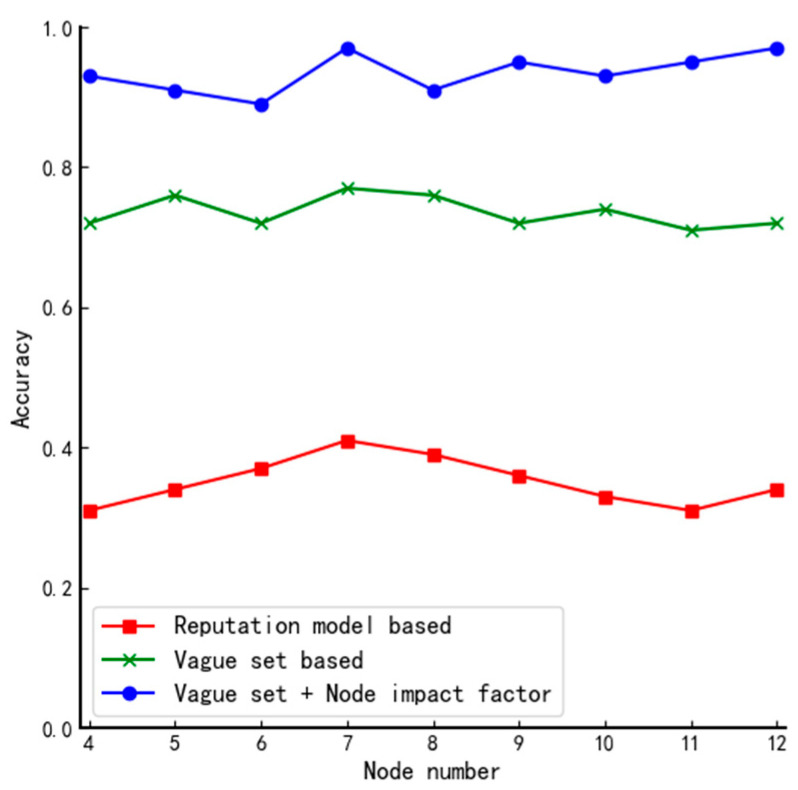
Comparison of accuracy of algorithms in complex structures.

**Figure 10 entropy-24-01013-f010:**
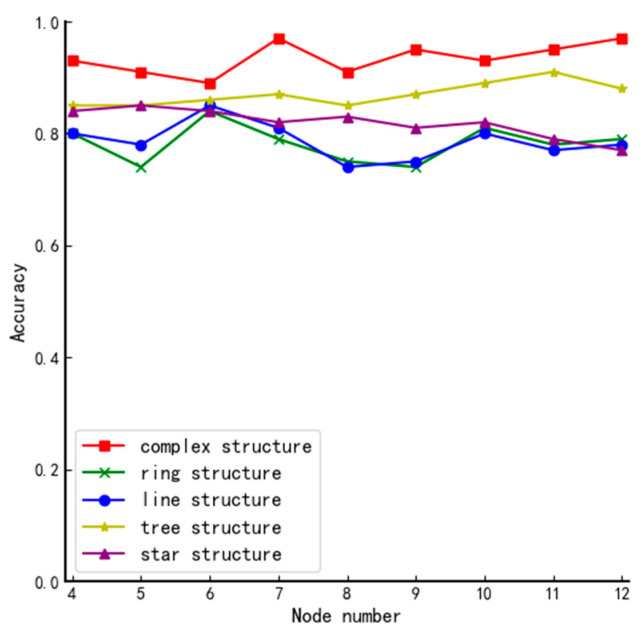
Comparison of accuracy in different structures.

**Table 1 entropy-24-01013-t001:** Vague value and fuzzy value results.

Number	Favor Vote	Abstention Vote	Against Vote	Vague Value	Fuzzy Value
2	6	4	1	[6/11, 10/11]	0.75862
7	5	5	1	[5/11, 10/11]	0.71429
6	3	6	2	[3/11, 9/11]	0.55556
9	3	6	2	[3/11, 9/11]	0.55556
5	4	2	5	[4/11, 6/11]	0.45161
3	3	4	4	[3/11, 7/11]	0.44828
4	3	3	5	[3/11, 6/11]	0.4
11	3	3	5	[3/11, 6/11]	0.4
12	3	3	5	[3/11, 6/11]	0.4
8	2	5	4	[2/11, 7/11]	0.39286
1	3	2	6	[3/11, 5/11]	0.35484
10	0	4	7	[0, 4/11]	0.13793

**Table 2 entropy-24-01013-t002:** First-order centrality of nodes.

Node (i)	1	2	3	4	5	6	7	8	9	10	11	12
$k_{i}$	1	1	4	5	4	4	1	4	3	2	4	1

**Table 3 entropy-24-01013-t003:** The impact factors of all nodes.

Node (i)	1	2	3	4	5	6	7	8	9	10	11	12
$I_{i}$	10.5	12	19	18	22	22	12	22	17	11.5	16	8.5

**Table 4 entropy-24-01013-t004:** Neighboring node voting statistics.

Number	First-Order Adjacency Node		Second-Order Adjacency Node	
	Favor	Against	Favor	Against
6	3, 11	5	12	4
9	10, 11	-	-	6

**Table 5 entropy-24-01013-t005:** Statistics of the vote share of alternate nodes.

Node Number	1	3	4	5	11	12
P	−17	−5	7	−15	−51	15

**Table 6 entropy-24-01013-t006:** Statistics on flat tickets for different size structures.

Number of Nodes	4	5	6	7	8	9	10
No tie-breaking	45	21	22	20	14	11	12
Two-node tie-breaking	38	70	48	37	37	29	27
Multi-node tie-breaking	17	9	30	43	49	60	61

## Data Availability

Not applicable.

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
