# Peer review of "Improvement of Delegated Proof of Stake Consensus Mechanism Based on Vague Set and Node Impact Factor"

_entropy, 2022, doi:10.3390/e24081013_

Round 1

Reviewer 1 Report

1. The authors claim that the performance of the algorithm in inherently linked to the complexity of the network structure. This claim needs more supports and has not been elaborately discussed in the paper. 

2. The paper also does not describe the process for when does a vote not qualify because every node selected with either vote for or against, or can even abstain from voting. How are traitor/unfavourable nodes identified in the system?

Author Response

Improvement of Delegated Proof of Stake Consensus Mecha-nism Based on Vague Set and Node Impact Factor

(Manuscript ID: entropy-1805624)

Response to Reviewer 1 Comments

Dear Reviewer:

Thank you for your letter and your suggestions on our manuscript. These comments are invaluable and help we recognize the shortcomings in the article and point us in the direction of improvement.

Here are the points you listed:

Point 1: The authors claim that the performance of the algorithm in inherently linked to the complexity of the network structure. This claim needs more supports and has not been elaborately discussed in the paper.

Point 2: The paper also does not describe the process for when does a vote not qualify because every node selected with either vote for or against, or can even abstain from voting. How are traitor/ unfavorable nodes identified in the system?

After careful consideration, we have re-adjusted the content of the article and summarized your recommendations into three aspects. Here are our responses:

Response 1: Thank you for your suggestion! We re-examined the experimental part and found that we only used one set of data from the beginning to the end for the experiments of the traditional algorithm, which lacked credibility. Therefore, we first modified and improved the results of some experimental graphs. Then, we rewrote the final experimental analysis. Compared with the previous one, we specifically analyzed the relationship between the algorithm proposed in this paper and the network complexity from the number of nodes and the network structure. Details are written on Line 418.

Response 2: Thank you for your suggestion! We have found that in the article we did not discuss possible special cases. Therefore, in this revision, we have added a description of possible special situations and how to deal with them. In addition, in general, the votes of all nodes will be voted according to the historical behavior of the nodes, and there may be malicious nodes in the system. Therefore, there is only a small probability that any node will receive only a single vote. It is also the same for the entire system. Therefore, when the voting types of all nodes in the system are the same, we need to check the security of the system. Details are written on Line 202.

 Please review our revised draft again, and we hope to receive your approval.

Best regards

Reviewer 2 Report

My main concerns about this paper are as follows:

  • The problem or reasoning of why the authors use "impact factor" and scores from "adjacent node" to calculate who wins the voting is not described.
  • Regardless of whether the node being overridden is malicious or not, if one node can influence the voting results (through the proposed impact factor scheme), does it not mean that the system is not fair?; taken from a democratic point of view.
  • The authors need to analyze their proposal's centralization and fairness since they claimed their proposal could solve the centralization tendency of the static DPoS (in line 12).

Other considerations:

  • Change the title to Title Case.
  • Line 38-39, consider changing the term to PoW-variant, PoS-variant, and BFT-variant.
  • Add a reference in the sentence "Larimer proposed DPoS....." (in line 49-51)
  • Line 57, security copper leakage?
  • Line 97-97, consider using the term "in favor votes" and "against votes"
  • Line 110, pf these parameters?
  • Line 142, ealgorithm?
  • Figure 2, how can the system moves from "T >= \delta T" to "t >= \delta t"? If yes, go to the left side; if no, go to the right side. On what condition do we move downward?
  • Line 154, What do the authors mean by "the article" here?
  • In Section 3.2, the first-order centrality is calculated based on Figure 2. Why do nodes in the middle then have a high impact factor? Why is the impact factor calculated based on the topology? In the blockchain, all nodes will sign the vote; therefore, no nodes can modify others' votes maliciously.
  • Figures 3, 4, 7, 8, and 9 must be enlarged.
  • Table 6, add "Number of nodes" on top of "4 5 6 7 8 9 10"
  • In figure 4, it is unclear what the author means by the rate on the y-axis. What kind of rate is this?
  • Line 324-326, "we can find that when the traditional DPoS voting method is..."; it is unclear what the traditional DPoS voting method is in Figure 3. Is it the red, green, or blue line?
  • Line 345, Figure 5?
  • Section 4.3.1 is "Simple Structure," but Figure 5 says "single structure"; which one is correct?
  • Line 224, Figure 3?
  • Line 239, Figure 3?
  • Figure 5, consider changing the caption to "... star (a), tree (b), circular (c), and linear (d) structures"
  • Why are the red and green lines in Figures 5 (a-d), 7 (a-b), and Figure 8 all the same?

Author Response

Improvement of Delegated Proof of Stake Consensus Mechanism Based on Vague Set and Node Impact Factor

(Manuscript ID: entropy-1805624)

Response to Reviewer 2 Comments

Dear Reviewer:

Thank you for your letter and your suggestions on our manuscript. These comments are invaluable and help we recognize the shortcomings in the article and point us in the direction of improvement.

Thank you again for your careful attention and we are very sorry for the many errors of detail in our article.

According to your suggestion, we briefly summarize it into two aspects: the discussion on fairness and the discussion on the node impact factor. In addition, some of the content in the experimental part has also been revised.

Here are the points you listed:

Point 1: The problem or reasoning of why the authors use "impact factor" and scores from "adjacent node" to calculate who wins the voting is not described.

Point 2: Regardless of whether the node being overridden is malicious or not, if one node can influence the voting results (through the proposed impact factor scheme), does it not mean that the system is not fair?; taken from a democratic point of view.

Point 3: The authors need to analyze their proposal's centralization and fairness since they claimed their proposal could solve the centralization tendency of the static DPoS (in line 12).

Point 4: Other considerations.

After careful consideration, we have re-adjusted the content of the article and summarized your recommendations into three aspects. Here are our responses:

Aspect 1: "impact factor" and scores analysis

Response 1: Thank you for your suggestion! We are sorry that our article did not accurately express our views. The method we propose contains two elections in total, in which the first round of elections is necessary, and in this round, all nodes have the same voting share, which is not affected by the node's own influence factor, so it can ensure fairness. The second round of elections is not inevitable. It only appeared when the first round of elections failed to yield a valid result. At this time, in order to better ensure the security of the final winning node, according to the principle of information transmission, it is necessary to comprehensively consider the influence factor of the node itself and the voting situation of its adjacent nodes. Although the second round of elections nominally introduces weights in the voting of each node, in fact the fairness of all nodes is not destroyed.

Aspect 2: centralization and fairness analysis

Response 2: Thank you for your suggestion! We have re-analyzed and evaluated the article and found that the algorithm proposed in this article ensures that all nodes have the same share of voting rights in the first stage of the election. When the first stage of the election fails to completely select the member nodes, the algorithm fully considers the criticality of nodes in different network structures, and uses this property to conduct the second round of screening. On the whole, fairness and democracy of the network can be guaranteed. When we were verifying the decentralized nature of the algorithm, we found that the experimental results did not achieve the theoretical results, and there was still a phenomenon of centralization. Based on this, we have deleted the description of decentralization in the abstract.

Aspect 3: Experiments

Response 3: Thank you for your suggestion! In the original experiment, we only conducted one simulation experiment on the traditional reputation value algorithm and the existing improved algorithm based on fuzzy sets. Therefore, there are cases where the red line and the green line are exactly the same in many pictures. After receiving your comments, we have found that this is not advisable and have experimented with supplements several times. We found that the experimental results are not completely consistent. The revised figure is shown in the experimental part of the article.

Please review our revised draft again, and we hope to receive your approval.

Best regards

Round 2

Reviewer 2 Report

The authors have addressed all of my comments.

The overall quality of the paper has been improved.

Please check the grammar and spelling one more time before publishing.